# Hematin- and Hemin-Induced Spherization and Hemolysis of Human Erythrocytes Are Independent of Extracellular Calcium Concentration

**DOI:** 10.3390/cells13060554

**Published:** 2024-03-21

**Authors:** Diana M. Mikhailova, Elisaveta Skverchinskaya, Julia Sudnitsyna, Kirill R. Butov, Ekaterina M. Koltsova, Igor V. Mindukshev, Stepan Gambaryan

**Affiliations:** 1Sechenov Institute of Evolutionary Physiology and Biochemistry, Russian Academy of Sciences, 44 Thorez Ave., 194223 Saint Petersburg, Russia; mikhailowa.dm@gmail.com (D.M.M.); lisarafail@mail.ru (E.S.); julia.sudnitsyna@gmail.com (J.S.); iv_mindukshev@mail.ru (I.V.M.); 2Department of Cytology and Histology, Saint Petersburg State University, 7/9 Universitetskaya Emb., 199034 Saint Petersburg, Russia; 3Department of Molecular Biology and Medical Biotechnology, Pirogov Russian National Research Medical University, 117997 Moscow, Russia; krbutov@gmail.com; 4Dmitry Rogachev National Medical Research Center of Pediatric Hematology, Oncology and Immunology, 117997 Moscow, Russia; ekaterina_koltsova@bk.ru; 5Center for Theoretical Problems of Physicochemical Pharmacology, Russian Academy of Sciences, 30 Srednyaya Kalitnikovskaya st., 109029 Moscow, Russia

**Keywords:** erythrocytes, hematin, hemin, cell spherization, hemolysis, albumin, laser diffraction

## Abstract

Pathologies such as malaria, hemorrhagic stroke, sickle cell disease, and thalassemia are characterized by the release of hemoglobin degradation products from damaged RBCs. Hematin (liganded with OH^−^) and hemin (liganded with Cl^−^)—are the oxidized forms of heme with toxic properties due to their hydrophobicity and the presence of redox-active Fe^3^. In the present study, using the original LaSca-TM laser particle analyzer, flow cytometry, and confocal microscopy, we showed that both hematin and hemin induce dose-dependent RBC spherization and hemolysis with ghost formation. Hematin and hemin at nanomolar concentrations increased [Ca^2+^]_i_ in RBC; however, spherization and hemolysis occurred in the presence and absence of calcium, indicating that both processes are independent of [Ca^2+^]_i_. Both compounds triggered acute phosphatidylserine exposure on the membrane surface, reversible after 60 min of incubation. A comparison of hematin and hemin effects on RBCs revealed that hematin is a more reactive toxic metabolite than hemin towards human RBCs. The toxic effects of heme derivatives were reduced and even reversed in the presence of albumin, indicating the presence in RBCs of the own recovery system against the toxic effects of heme derivatives.

## 1. Introduction

Free heme, ferrous (Fe^2+^) protoporphyrin (PP) IX, and heme derivatives are widely recognized pathological molecules in a number of hemolytic conditions, and their high concentrations (up to 500 µM) emerge in the human organism during malaria [1,2], hemorrhagic stroke [3,4,5], gastric ulcer [6], sickle cell disease, and beta-thalassemia [7]. Moreover, accumulation of degraded hemoglobin in stored RBCs can also lead to hematin/hemin formation, and the supernatant of such blood units was shown to trigger inflammatory activation of neutrophils, meaning that transfusion of such units might contribute to transfusion effectiveness decrease and in the worst cases to poor clinical outcomes [8,9]. Hematin (ferric (Fe^3+^) PP IX liganded with OH^−^) is formed predominantly during malaria RBC invasion as a by-product of the parasite’s digestion of host cell hemoglobin (Hb), while hemin (Fe^3+^ PP IX liganded with Cl^−^) is constantly produced in the human body during the reactions of Hb autooxidation either spontaneous or disease-triggered (Appendix A) [10]. Both compounds were shown to induce complications such as vascular occlusion, pulmonary damage, secondary brain injury, and kidney or liver damage [11,12,13,14].

Hemin and hematin are able to activate endothelial cells, triggering endothelium dysfunction [15]; induce ferroptosis of platelets [16], damage neurons and astrocytes [5,17], and cause hemolysis of red blood cells (RBCs) [18,19]. At the same time, hemin supplements have been established as the standard of care for the treatment of acute attacks of porphyria. Thus, Panhematin (hemin black powder) and Normosang (complex of hemin, arginine, and ethanol) are effectively used in clinical practice [20]. However, such treatment is associated with hemostasis-affecting side effects, which include phlebitis and coagulopathies characterized by thrombocytopenia [21].

RBCs have a high affinity for heme derivatives [22]. Being hydrophobic molecules, hematin and hemin intercalate into the cell membrane, resulting in disruption of the erythrocyte membrane skeleton, weakening the junctions between spectrin and band 4.1, finally leading to hemolysis [23]. The colloid-osmotic mechanism can explain hemolysis: hemin triggers the loss of potassium from RBCs, activating calcium-sensitive potassium channels [24]. However, whether hematin- and hemin-induced transformation depend on [Ca^2+]^_i_ is unknown. Furthermore, an electron microscopy study has established that hemin induced the transformation of RBCs to echinocytes at the lower concentrations, while the higher concentrations led to the transformation to spherocytes [25]. Moreover, it has been shown that hemin induced the formation of specific topological structures called ‘grains’ on the surface of RBCs and stimulated eryptosis with typical hallmarks such as membrane scrambling, cell shrinkage, and phosphatidylserine (PS)-exposure on the erythrocyte surface [19].

In literature, hematin and hemin are generally designated as ‘hemin’ without indicating their differences. However, these compounds have different structures due to their ligands, OH^−^ in hematin and Cl^−^ in hemin (Appendix A) [10]. Both compounds are released in blood during pathologies, but no concrete data exist on whether they have similar effects on RBCs. Thus, one of the main aims of this study was to analyze hematin and hemin individually and compare their effects on RBCs. In addition, as the distinct mechanism of heme derivatives’ action on erythrocytes is still unclear, it is crucial to expand the available data, especially considering the methodology of hemin/hematin-based medicines infusion, i.e., directly into the bloodstream [26]. Whether hematin and hemin induce the acute increase in intracellular Ca^2+^ concentration ([Ca^2+^]_i_) also remains unknown. In addition, there are no data on whether hematin- or hemin-induced transformation of RBCs is reversible and whether albumin could contribute to erythrocyte recovery to their normal “biconcave” shape. These data will help to understand the mechanism of hematin and/or hemin effects on erythrocytes in different pathological situations.

In this study, we showed that hematin/hemin in nanomolar concentrations induced dose-dependent spherization of RBCs, whereas increased concentrations (up to 5 µM) triggered hemolysis of RBC, which finally led to ghost formation. Hematin/hemin at nanomolar concentrations increased [Ca^2+^]_i_ in RBC; however, spherization and hemolysis occurred either in the presence or absence of calcium, indicating that both processes are independent of [Ca^2+^]_i._ We showed that hematin effects on RBCs were more pronounced than those of hemin, indicating that hematin is a more potent agonist towards human RBC transformation. Notably, albumin not only decreased hematin/hemin-induced transformation of RBCs but also participated in the return of “swollen” spherical RBCs to their native biconcave discoid shape.

## 2. Materials and Methods

### 2.1. Reagents and Working Buffers

Hemin, albumin, calcium ionophore A23187, tert-butyl hydroperoxide (t-BuOOH), Calcein-AM (C-AM), and buffer components (HEPES, NaCl, KCl, MgCl_2_, d-glucose, EGTA, CaCl_2_) were purchased from Sigma-Aldrich (Darmstadt, Germany). Fluo-3-AM and Annexin-APC were obtained from Invitrogen (Carlsbad, CA, USA). Lactadherin-FITC was synthesized in Dmitry Rogachev National Medical Research Center of Pediatric Hematology, Oncology, and Immunology, and a detailed description is presented in the Appendix A [27].

HEPES buffer was adapted for the work with human RBCs and contained in mM: NaCl, 140; HEPES, 10; KCl, 2; MgCl_2_, 2.5; D-glucose, 5; EGTA, 2; pH 7.4 (pH meter FiveEasy F30, Mettler Toledo, Columbus, OH, USA). The osmolality of the buffer (300 mOsm/kg H_2_O) was controlled by the cryoscopic osmometer Osmomat 030 (Gonotec GmbH, Berlin, Germany).

### 2.2. RBC Preparation

Human blood was drawn from healthy donors after signing a written informed consent and was performed according to the Declaration of Helsinki and approved by the Ethics Committee of the Sechenov Institute of Evolutionary Physiology and Biochemistry of the Russian Academy of Sciences (protocol no. 1–04 from 7 April 2022). Venous blood was collected in S-monovette tubes (9NC, Sarstedt, Nümbrecht, Germany) with 2 mM EGTA and centrifuged at 400× *g* (centrifuge ELMI-50CM, Elmi, Riga, Latvia) for 3 min. RBCs were washed twice in HEPES buffer and were diluted to 0.1 × 10^9^ cells/mL working concentration.

### 2.3. Hematin and Hemin Preparation

To prepare the hemin stock solution (1 mM), hemin powder was dissolved in DMSO. For the preparation of hematin stock solution (1 mM), hemin powder was dissolved in 20 mM NaOH. The maximum absorbance (λ max) for the two molecules is different, around 370 nm for hemin chloride and around 383 and 400 nm for hematin [28]. Hematin and hemin concentrations were controlled using the molar extinction method according to the Beer–Lambert equation using the compounds’ millimolar extinction coefficient at two different wavelengths. For hematin, we used ε = 58.4 mM^−1^cm^−1^ at λ = 385 nm, described previously in [7]. It should be mentioned that in the paper we quoted before [7], hematin was indicated as hemin. Similarly, using the data on hematin spectra, we determined ε for hemin at the wavelength λ = 342 nm, and ε was 59.5 mM^−1^cm^−1^. Spectra were registered using a spectrophotometer (Spectroscopic Systems LTD, Moscow, Russia), and the representative spectra are presented in Appendix A.

### 2.4. Analysis of Hematin and Hemin Effects on Human RBCs by the Laser Diffraction Method

Hemin- and hematin-induced transformation of RBCs was registered by the laser diffraction method (laser microparticle analyzer LaSca-TM, BioMedSystems Ltd., Saint Petersburg, Russia), adapted for cell physiology and used according to [29]. The method is characterized in detail in [30]. Shortly, the laser beam (650 nm) passed through the cuvette with continuous stirring (1200 rpm, 37 °C) with RBCs suspension in HEPES buffer (10^6^ cell/mL final concentration). The light scattering intensity (LSI) was continuously detected by photodiodes at the angles from 1° to 12°. In our experiments, the angles of 2.5° and 12° were used to characterize cell shape transformations, as these angles are the most informative on cell shape changes according to the light scattering indicatrix of human RBCs, described in Appendix A. Spherization of RBCs was characterized by the decrease of light scattering intensity at 2.5° and an increase at 12°. The methodology of RBC shape change analysis by laser analyzer is described in [31]. Spherization of RBCs corresponded to a lower amplitude of the LSI signal compared to control. Hemolysis of RBCs corresponded to the decrease in light scattering intensity signal in all angles.

The following parameters of RBC transformation were evaluated:Spherization index—the control LSI oscillation amplitude ratio to LSI oscillation amplitude after hematin/hemin addition. This index can be used to distinguish between the normal “biconcave disk” form and spherical shapes of dysmorphic RBCs. LSI oscillation amplitude for the control was taken as a spherization index of zero.The rates of spherization and hemolysis—the dynamics of LSI signal change over time during the process of interestPercent of hemolysis (% hemolysis)—is the ratio of minimum LSI signal to maximum.

The detailed characterization of RBC transformation by laser diffraction is presented in the Appendix A.

### 2.5. Intracellular Calcium Concentration Determination

To monitor the changes in [Ca^2+^]_i_ induced by hematin/hemin, the upgraded laser microparticle analyzer LaSca-TM with built-in 488 nm laser and fluorescence detector of 527 nm (FL1) (LaSca-TMF, BioMedSystems Ltd., Saint Petersburg, Russia) suitable for kinetic fluorescence measurement was used. This device allows the analysis of RBC transformation based on the LSI signal changes simultaneously with the changes in intracellular calcium concentration determination according to the Fluo-3 fluorescence signal. Previously, LaSca-TMF was used to evaluate the [Ca^2+^]_i_ in platelets [32]. Here, we adapted this method for RBCs. Washed erythrocytes (10^9^ cells/mL) were incubated with Fluo-3-AM (10 µM, 60 min, 37 °C) in the dark. Then, RBCs were suspended in HEPES buffer (10^6^ cells/mL, final concentration) and analyzed by both fluorescence and laser diffraction analysis. For [Ca^2+^]_i_ concentration calibration in Fluo-3-stained RBCs, we used calcium ionophore A23187 to achieve the maximal fluorescence signal for Fluo-3 and EGTA—for the minimum Fluo-3 signal. [Ca^2+^]_i_ was then calculated using the equation presented in Appendix A.

### 2.6. Flow Cytometry Analysis

Hematin- and hemin-induced transformation of RBCs was analyzed by flow cytometry using the CytoFLEX flow cytometer (Beckman Coulter, Brea, CA, USA) with an analysis of not less than 10,000 events.

#### 2.6.1. Characterization of RBC Distribution by Sizes, Complexity, and Granularity

RBCs (10^6^ cells/mL) diluted in HEPES buffer corresponded to control and were characterized by a wide distribution of events on a density plot. After hematin/hemin addition to the suspended RBCs, the density plot narrowed, i.e., the cells became more homogenous in size or volume, corresponding to the spherization of RBCs. For the analysis of hematin/hemin-induced transformation of RBCs, we used forward scatter (FSC)/side scatter (SSC) mode. Forward scatter corresponds to the cell size, side scatter correlates with internal complexity and granularity [33].

#### 2.6.2. Phosphatidylserine (PS) Externalization Test

To detect the PS exposure on the RBC outer membrane, we used two different compounds, Annexin and Lactadherin, because the latter is supposed to be an earlier marker of apoptosis in contrast to annexin [34]. RBC (10^6^ cells/mL) were incubated with hematin/hemin (15, 30, 60 min, 1 μM) and then stained with Lactadherin-FITC or Annexin-APC (15 min, RT, according to the manufacturer recommendations). Lactadherin was registered at fluorescence light sensor 1 (FL1, FITC), annexin—at FL4, APC.

#### 2.6.3. Cell Viability Test

Calcein-AM (C-AM) was used to determine whether hematin and hemin affect cell viability. RBCs (10^6^ cells/mL) were incubated with C-AM (0.5 µM, 40 min, 37 °C), then hematin or hemin at indicated concentrations were added to the cells, and the calcein fluorescence was analyzed at FL1.

### 2.7. Fluorescence and Bright Field Microscopy

Hematin- and hemin-induced transformation of RBCs was analyzed by Leica TCS SP5 MP scanning confocal microscope (Leica Microsystems Inc., Bannockburn, IL, USA) with 20× and 63× immersion objectives. Washed RBCs suspended in HEPES buffer (10^6^ cells/mL) were placed in the plastic dish for confocal microscopy and analyzed in the bright field for shape change analysis and in the FITC channel for cell vitality characterization. Hematin and hemin at indicated concentrations were added to the cell suspension in the dish, and RBC transformation was registered in kinetics. The figures were also additionally zoomed in for better visualization.

### 2.8. Data Analysis

Flow cytometry data were analyzed using the original software CytExpert (BeckmanCoulter, Brea, CA, USA). Confocal microscopy data were analyzed using the Leica TCS SPII confocal software (Leica Microsystems Heidelberg GmbH, Heidelberg, Germany). Data obtained by the laser diffraction method were analyzed using the original software LaSca_32 v.1498 (BioMedSystems Ltd., Saint Petersburg, Russia) of the laser particle analyzer LaSca-TM. Statistical analysis and calculation of EC50 were performed in GraphPad Prism v.9 (GraphPad Software Inc., San Diego, CA, USA). According to the Kolmogorov–Smirnov test, the data were normally distributed; thus, the differences between the two groups were compared using an unpaired *t*-test. For multiple comparisons, one-way ANOVA followed by Tukey post hoc was used. Data are presented as mean ± SD. Each set of experiments was performed at least five times (*n* = 5), and *p* < 0.05 was considered statistically significant.

## 3. Results

### 3.1. Hematin Induces Transformation of RBCs

As described, hematin and hemin are related heme derivatives released in the blood during various diseases [5,7,11,35,36], but they have different ligands attached to Fe^3+^ [10], which can potentially lead to different effects on RBCs. Hemin was previously shown to trigger the formation of dysmorphic RBCs with grain-like structures on the membrane surface [37]. Therefore, we analyzed the effects of the other heme derivative, hematin, on RBC morphology using fluorescent microscopy.

Hematin at low concentrations (up to 1 µM) triggered RBC transformation to echinocytes with the formation of specific granules on the membrane surface. At higher concentrations (up to 2 µM), RBCs underwent spherization, thus, the cell size visually (diameter-wise) decreased (Figure 1, 2 µM). However, the fluorescence of calcein did not decrease, indicating that cell viability was maintained (Figure 1, last column). At the highest tested concentration of hematin (5 µM), we observed RBC lysis with Hb release and ghost formation (Figure 1, 5 µM) accompanied by a significant calcein fluorescence decrease.

Next, we analyzed the effects of hematin using flow cytometry. Hematin at indicated concentrations was added to RBC suspension, and the effects on cell vitality were analyzed based on calcein fluorescence changes. Hematin at 1 µM led to a decrease of calcein positive events count (Figure 2, 1 µM). At the highest concentration (5 µM), the fluorescence of calcein and events count (i.e., cell number) decreased (Figure 2, 5 µM), confirming the results observed earlier using microscopy: cell lysis and ghost formation (Figure 1, hematin 5 µM).

Next, we characterized the effects of hematin on RBC transformation in the axis of cell size distribution pattern using flow cytometry. Hematin triggered the RBC distribution width contraction, which corresponded to the spherization of RBCs in a dose-dependent manner (Figure 3). Hematin at a concentration of 5 μM triggered almost complete hemolysis, corresponding to the previously obtained data (Figure 1 and Figure 2). To test whether hematin/hemin could provoke microparticle (MP) formation, we used t-BuOOH as a positive control of MP formation (Figure 3, t-BuOOH, yellow gate). As was shown previously, oxidative stress induced by t-BuOOH leads to MP formation in human RBCs [38]. However, even at a high concentration, hematin did not induce MP formation in contrast to t-BuOOH (Figure 3, framed in red, yellow gate). To prove that the signal detected by flow cytometry after lysis of RBCs induced by 5 µM of hematin is a free Hb, we lysed washed RBCs in water and analyzed collected supernatant containing only free Hb by flow cytometry (Appendix A). These results supported our hypothesis that during hematin/hemin-induced hemolysis, the free Hb, and not MPs, is released from RBCs and turned them into erythrocyte ghosts.

### 3.2. Hematin and Hemin Trigger Spherization and Hemolysis in a Dose-Dependent Manner

Next, to fully characterize RBC transformation induced by heme derivatives, we analyzed the hematin and hemin effects of RBCs in kinetics using laser diffraction. Hematin- and hemin-induced cell transformations were characterized based on laser scattering intensity changes at 2.5°and 12°. A detailed description of cell transformation analysis using laser diffraction is presented in Appendix A.

#### 3.2.1. Hematin and Hemin Induce Spherization of RBCs

Hemin and hematin, both at low concentrations, led to RBC transformation from normocytes to dysmorphic cells. Therefore, here we analyzed the RBC spherization triggered by both agonists. To analyze agonist effects, we used (a) spherization index and (b) spherization rate parameters. Spherization of RBCs was observed starting from very low doses of hematin (30 nM) and hemin (100 nM), and maximal spherization indices were reached at 1000 nM and 1500 nM, respectively (Figure 4A). The spherization index was plotted against the concentration of the effectors, and the EC50 was calculated (Figure 4B). EC50 of hematin was significantly lower than that of hemin (Figure 4C), indicating that hematin is a more reactive agonist of RBC spherization than hemin.

Spherization rate is another important parameter to characterize hematin- and hemin-induced transformation of RBCs, which is calculated as the rate of LSI increase measured at 12° (for the details, see Appendix A). The spherization rates were plotted against hematin and hemin concentrations (Figure 4D). Maximal rates of spherization (V_max_) before hemolysis were 0.482 ± 0.072 and 0.174 ± 0.028 for hematin and hemin, respectively (Figure 4E). These data clearly indicate that hematin is a more reactive compound than hemin towards RBC transformation.

#### 3.2.2. Hematin and Hemin Trigger Hemolysis of RBCs

At high concentrations (10–50 µM), hematin induces RBC lysis [18]. However, there are no data concerning minimal concentrations triggering hemolysis. In addition, it is unknown whether there are any differences in hemolysis induced by these two heme derivatives. Thus, we analyzed the hematin- and hemin-induced RBC lysis in detail using the following parameters: hemolysis % and hemolysis rate (for the details, see Appendix A).

RBC lysis was observed when agonist concentrations that led to spherization were exceeded, i.e., starting from 1.5 μM and 2 μM for hematin and hemin, respectively (Figure 5A). To calculate EC50, hemolysis % was plotted against agonist concentrations (Figure 5B,C). The EC50 for hematin turned out to be lower than that of hemin (Figure 5C), which is consistent with all previously described data.

Hemolysis rate (V_hem_) characterizes the rate of the membrane and skeleton disassembly; therefore, we determined it for the lysis triggered by hematin and by hemin (Figure 5D) and analyzed the maximal hemolysis rates (V_max_). The V_max_ for hematin was higher than that of hemin (Figure 5E).

These data, along with RBC spherization data, confirm the higher reactivity, or toxicity, of hematin compared to hemin towards human RBCs, indicating again that hematin is a more potent agonist for both processes, RBC transformation along with RBC lysis.

### 3.3. Hematin- and Hemin-Induced RBC Spherization Is Accompanied by a Rise in Intracellular Calcium Concentration

Hemin addition to RBCs was shown to trigger the increase in the cytosolic calcium concentration in RBCs after long-term incubation (10 µM final concentration, 48 h) [18]. Here, we analyzed whether hematin or hemin would induce the increase in [Ca^2+^]_i_ at low concentrations of agonist, which induce spherization.

Hematin- and hemin-induced cell transformations were controlled by laser diffraction based on LSI changes at 12°, and corresponding changes in [Ca^2+^]_i_ were registered at the FL1 and characterized according to Fluo-3 fluorescence changes (Figure 6A). Calcium ionophore (A23187) and calcium chelator (EGTA) were used to calibrate the [Ca^2+^]_i_ in RBCs (Appendix A).

Both hematin and hemin at indicated concentrations induced an acute increase in [Ca^2+^]_i_ (Figure 6B). Hematin induced an increase in [Ca^2+^]_i_ with a trend to be slightly higher than that of hemin, but not significant.

### 3.4. Hematin Induces Exposure of PS on the RBC Surface

Hemin (1–10 µM) induced PS exposure on the RBC membrane after 48 h of incubation [17]. However, it is not known whether hemin or hematin would induce acute PS exposure. For PS exposure analysis, we used Annexin-APC and Lactadherin-FITC, which was shown to be a more sensitive agent for PS detection and could be used as an earlier marker of cell apoptosis [34,39]. A detailed description of the cloning and purification of Lactadherin is presented in Appendix A.

Hematin in concentration that triggered spherization (1 µM) was added to the RBCs (10^6^ cells/mL), and co-incubation with agonist lasted for 15, 30, and 60 min. Then, the cells were stained with either Lactadherin-FITC or Annexin-APC. For the positive control of PS exposure, Ca^2+^ ionophore A23187 was used. After 15 min of incubation with hematin, the percent of Annexin- and Lactadherin-positive events was almost equal, with no significant differences (Figure 7A,B and Figure 8). After 30 min of incubation with the agonist, the percentage of Lactadherin-positive events was detected as maximal while the Annexin-positive events count started to decrease (Figure 7A,B and Figure 8). After 60 min of incubation, both Lactadherin- and Annexin-positive evens amounts decreased (Figure 7A,B and Figure 8). It is essential to mention that in the SSC/FSC axis, the corresponding RBC distribution width increased over the incubation time, with the agonist returning almost to that of control cells (Figure 7C). These data indicated that hematin (1 µM) induced an acute PS exposure in human RBCs with the maximal effect after 30 min incubation with an agonist. Taken together, data on PS exposure and cell distribution width indicate the possible existence of self-recovery mechanisms in human RBCs towards hematin toxicity. Additionally, we showed that Lactadherin is a more sensitive dye for PS exposure determination. Hemin-induced PS exposure was similar to that of hematin, though the effect was not that pronounced as in case of hematin.

### 3.5. Hematin- and Hemin-Induced Spherization of RBCs Is Decreased in the Presence of Albumin

Albumin, being the most abundant plasma protein, could bind heme derivatives with a high affinity, thus suppressing their catalytic and peroxidative effects. Moreover, albumin is capable of forming complexes with heme at a quantification ratio of 1:1 [40]. The albumin concentration in the bloodstream is 400–600 µM, drastically exceeding hematin/hemin concentration [41]. Nevertheless, in hemolytic disorders, local concentrations of heme derivatives may be significantly higher than albumin concentrations, for instance, in hematoma during hemorrhagic stroke [5]. Thus, the next step of our study was to find the concentration ratio between albumin and heme derivatives of interest and to compare the suppressive effects of albumin on hematin and hemin.

To characterize hematin- and hemin-induced RBC transformations, we used a modified HEPES buffer with albumin addition in a range (1.5–15–75 μM). Experiments in the absence of albumin were taken as a positive control (Figure 9A,B). Thus, the maximal spherization index of RBCs was induced by 0.7 μM of hematin and 1 μM of hemin, and the corresponding kinetic parameters were taken as a control. In the presence of albumin (1.5 μM), the agonist concentration required for RBC spherization increased to 10 μM and 20 μM for hematin and hemin, respectively (Figure 9). In the presence of albumin in a higher concentration (15 μM), the maximal spherization index was observed at a concentration of 50 μM for both agonists. Interestingly, the spherization index increase triggered by hemin in the presence of albumin was again significantly lower than that of hematin. The presence of albumin at a concentration of 75 μM led to complete inhibition of both hematin- and hemin-induced transformation of RBCs.

Based on these data, the spherization indices were plotted against the agonist concentration, and the EC50 values were determined (Figure 10A). In the presence of albumin, the dose-dependency curves shifted to the right along the x-axis, indicating an increase in EC50 required to reach the effect (Figure 10A,B). In the presence of albumin, hemin-induced RBC spherization was inhibited in a more pronounced way than that for hematin. However, at the lowest tested albumin concentration (1.5 µM), the difference between the effects triggered by both compounds was not significant. Similarly, albumin also significantly decreased the spherization rate triggered by hematin and hemin (Figure 10C,D).

According to albumin concentration and EC50 of hematin/hemin in the presence of albumin concentration ratio (hemin, hematin EC50/albumin, µM) were calculated (Table 1). Albumin bound hemin in a higher concentration ratio in contrast to hematin.

### 3.6. Dysmorphic RBCs Return to the Biconcave Shape in the Presence of Albumin

Albumin is able to draw off hemin trapped in the RBC membrane and is capable of reversing storage-induced transformation to echinocytes [7,42,43]. However, it remains unclear whether RBCs can return from hemin- and hematin-triggered spherized state to their native shape in the presence of albumin.

To analyze this, we added albumin in a range of concentrations (0.5–2 μM) to the transformed cells and characterized the response by laser diffraction and flow cytometry (Figure 11). Hematin (0.5 μM) was added to RBCs to induce spherization, and then albumin (1 μM, data not shown) was added to the suspension, but it did not prevent spherization. At the same time, in the presence of albumin at a higher concentration (2 µM), RBCs were able to restore the normal biconcave shape from dysmorphic. These data were confirmed by flow cytometry (Figure 11B) and fluorescent microscopy (Figure 11C).

### 3.7. Hemin- and Hematin-Induced Hemolysis Is Inhibited in the Presence of Albumin

Albumin inhibited hematin/hemin-induced spherization of RBCs and could participate in their return to the normal biconcave form. However, the concentrations of albumin that could prevent RBC hemolysis are unknown. Therefore, we characterized hematin- and hemin-induced hemolysis in the presence of different albumin concentrations.

We used only 1.5 μM of albumin because the higher albumin concentration completely inhibited hemolysis of RBCs in the range of 1–50 μM of hematin/hemin. In addition, 1.5 μM of albumin completely prevented hemolysis of RBCs induced by hemin. Thus, we analyzed only hematin-triggered hemolysis in these experiments. Experiments without albumin addition were used for positive control where 5 μM of hematin induced 100% hemolysis. When 1.5 μM of albumin was added to the cells, the EC50 of hematin was significantly increased (Figure 12B,C), whereas the hemolysis rate decreased (Figure 12D,E), indicating that even low albumin concentrations significantly decreased RBC hemolysis.

### 3.8. Hematin-Induced RBCs Transformation Is Ca^2+^-Independent

Intracellular calcium regulates a number of processes in RBCs, including maintaining the normal discoid shape control of membrane lipid composition and permeability [44]. To determine the role of Ca^2+^ in the hematin-induced transformation of RBCs, we compared hematin-induced cell transformation in the presence and absence of extracellular Ca^2+^. Hematin at indicated concentrations was added to RBCs (10^6^ cells/mL) diluted in HEPES buffer with EGTA (2 mM) or Ca^2+^ (1.5 mM) and analyzed cellular response by flow cytometry (Figure 13).

The pattern of RBC distribution width and the number of events did not differ in the Ca^2+^-containing HEPES buffer compared to EGTA-containing HEPES buffer, indicating that hematin-induced RBC transformation was Ca^2+^-independent.

## 4. Discussion

At basal conditions, the concentration of free hemin/hematin in the bloodstream is maintained at a very low level, 0.1–1 μM [45]. However, during extravascular and intravascular disorders [2,5,7,35,46,47] heme derivatives could be released from erythrocytes at a high concentration of up to 500 μM [48]. Hematin and hemin are hydrophobic molecules and, thus, are able to intercalate in the cell membrane, inducing changes in permeability and disruption of the membrane [18,49,50]. Furthermore, these compounds trigger the generation of reactive oxygen species (ROS) due to redox-active Fe atoms. All these toxic effects of hematin and hemin may lead to severe complications such as vascular occlusion, secondary brain injury, kidney and liver damage, and endothelium dysfunction [11,51,52,53].

Hematin and hemin both lead to RBC transformation, and it can be critical for blood rheology due to impeded red blood cell movement through capillaries [18,37]. Spherized RBCs lose their crucial ability to reversible deformation and become more sensitive to oxidative stress during different pathological conditions associated with high concentrations of hematin and hemin. The mechanism of hemin/hematin RBC spherization is still unclear, but the possible transformation scenario can be related to the disruption of junctions between transmembrane proteins and the membrane skeleton. It was shown that hemin alters the conformation of spectrin and possibly band 4.1, reduces spectrin self-associations as well as spectrin–protein 4.1 interaction, and finally decreases the stability of the RBC membrane [54]. In addition, hemin induces membrane shrinkage due to the activation of Ca^2+^-sensitive potassium channels and the efflux of KCl with osmotically bound water [55].

At the same time, in the literature, both compounds hematin and hemin are denoted as ‘hemin’, and there are no data on their differences in pathologies. Moreover, it is unknown which particular diseases are associated with either hemin or hematin formation and in what concentrations. Thus, one of our study’s aims was to compare these compounds’ effects on human RBCs to expand the existing knowledge. We have shown that both hematin- and hemin-induced dose-dependent spherization of RBCs start from very low concentrations (Figure 4). Nevertheless, in all the tested conditions, hematin showed higher reactivity towards human RBCs than hemin, which can be related to different ligands attached to Fe^3+^.

RBC transformation strongly depends on [Ca^2+^]_i_ [44], and previously, it was shown that hemin triggered an increase in [Ca^2+^]_i_ in RBCs during 48 h of incubation [18]. However, it is not known whether hematin/hemin would induce an acute increase of [Ca^2+^]_i_ and whether hematin/hemin-triggered transformation of RBCs would be Ca^2+^-dependent. Our experiments demonstrated that both compounds acutely increased [Ca^2+^]_i_ in RBCs, but spherization and hemolysis of RBCs also occurred in the media without calcium. These data clearly indicate that hematin- and hemin-induced increases in [Ca^2+^]_i_ and RBC transformation are independent processes.

Hemin induces hemolysis of RBCs in concentrations exceeding doses that cause spherization [19,22,24,36]. Hemolysis can be characterized by the colloid osmotic mechanism due to the increase in [Ca^2+^]_i_ [24]. However, our experiments clearly showed that spherization and hemolysis of RBCs occurred in both the absence and presence of Ca^2+^, indicating that hematin- and hemin-induced transformation was independent of [Ca^2+^]_i_. Notably, the process of RBC spherization started earlier than the increase in [Ca^2+^]_i_. Moreover, hemolysis was accompanied by Hb release (Appendix A) and ghost formation (Figure 1) without microparticle formation. Free Hb, in turn, is a toxic molecule [56] that may further complicate the progression of the above-mentioned diseases.

It is also known that hematin/hemin leads to membrane scrambling, where PS is exposed at the surface of the RBC membrane [18]. PS-exposing cells bind to PS receptors of macrophages and can be eliminated from circulation. In our experiments, we used not only traditional PS exposure marker Annexin but also Lactadherin, which turned out to be more sensitive to PS and can be used as an earlier marker of cell apoptosis [34,39]. We demonstrated hematin- and hemin-induced PS exposure on the RBC surface, but the fluorescence of Annexin and Lactadherin decreased over time (1 h) along with RBCs returning to their normal biconcave shape, indicating that hematin/hemin-induced PS exposure is reversible. In addition, the viability of spherized erythrocytes was not reduced according to the cell viability test. This suggests that erythrocytes might have their inner recovery system against the toxic effects of hemin/hematin. Nevertheless, surface exposure of PS can still compromise the microcirculation and may be highly procoagulant, inducing thrombotic complications.

We demonstrated that albumin effectively scavenged heme derivatives in a quantitative ratio of over 1:1/1:2, reducing spherization and hemolysis of RBCs (Table 1). Surprisingly, albumin can not only attenuate heme derivatives from the erythrocyte membrane but also contribute to the return of transformed cells to their standard biconcave shape. Despite the high affinity of albumin to heme derivatives and its high concentration in plasma, hematin and hemin can exhibit their toxic effects in pathological conditions [57]. For example, after a hemorrhagic stroke, the concentration of hematin/hemin in hematomas can be extremely high and exceed the local concentration of scavenging proteins such as albumin [5]. Furthermore, in severe complications such as sepsis, the level of serum albumin dramatically decreases due to increased leakage into the interstitial space because of possible hemin-induced increased capillary permeability [58], accelerated catabolism, and reduced hepatic synthesis of albumin resulting in toxic effects of free hematin and hemin [59].

## 5. Conclusions

Here, we showed that hematin and hemin induce dose-dependent spherization and hemolysis of RBCs accompanied by Hb release and ghost formation. Comparing two of these compounds revealed that hematin is a more reactive toxic metabolite than hemin. Both, hematin and hemin, induced PS exposure on RBC surface and caused an acute increase of [Ca^2+^]_i_. Nevertheless, Ca^2+^ did not affect either hematin- or hemin-triggered spherization and hemolysis, indicating that RBC transformation is Ca^2+^-independent. Importantly, spherization and PS exposure were reversible over time, and after 1 h of incubation, RBCs returned to their native forms. In addition, albumin not only reduced the toxic effects of heme derivatives but also participated in the return of already-transformed RBCs to their normal discoid form.

### 5.1. Future Research Directions and Clinical Applications

A future research aim is to investigate the molecular mechanism of hematin/hemin effects on human erythrocytes. Our future experiments are related to revealing which mechanisms are responsible for reversible spherization induced by low doses of hemin, what triggered RBC ghost formation, and which mechanisms are responsible for calcium-independent spherization and lysis. Understanding the binding sites of hematin and hemin to the erythrocyte membrane or the underlying skeleton would be a significant step in adjusting the therapy. Overall, such data could contribute to reducing the side effects of hemin injections and eventually result in better clinical outcomes, i.e., improving the patients’ quality of life.

### 5.2. Limitations of the Study

In our experiments, only washed RBCs were used, in rather dilute (10^6^ cells/mL) concentrations vs. real. Working with higher RBC concentrations was problematic for further analysis using either the LaSca particle analyzer or flow cytometer. In addition, the albumin concentration was significantly lower than in human plasma, and other defense mechanisms, such as GSH, should be considered. Thus, we cannot yet compare our data with real in vivo conditions.

## Figures and Tables

**Figure 1 cells-13-00554-f001:**
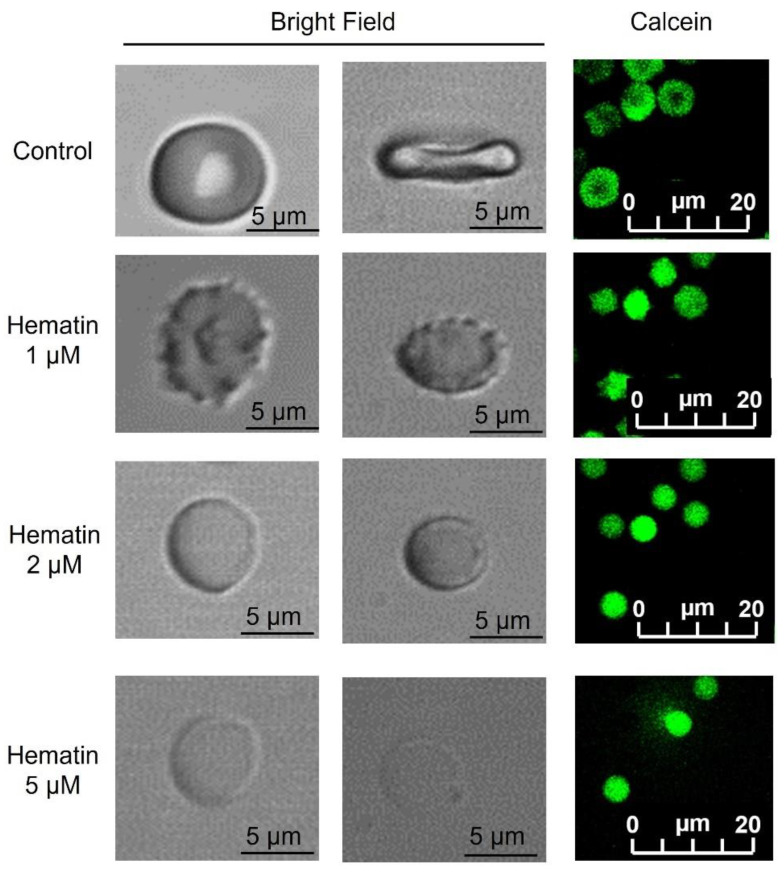
Hematin triggered the contortion of RBCs in a dose-dependent manner. Hematin at indicated concentrations was added to RBCs (10^6^ cells/mL) suspended in HEPES buffer without Ca^2+^, and the transformation of RBCs was characterized by confocal microscopy. Hematin induced the transition of RBCs from normal discoid form (control) to echinocytes (hematin, 1 µM) and then to spherical form (hematin, 2 µM) in a dose-dependent manner. Finally, erythrocytes lysed with further ghost formation. RBCs stained with Calcein-AM (0.5 µM, 40 min, 37 °C) were additionally visualized (last column, calcein, green). Representative images of hematin-transformed RBCs are shown. The first column of the bright field section is the view from above, and the second is the side view.

**Figure 2 cells-13-00554-f002:**
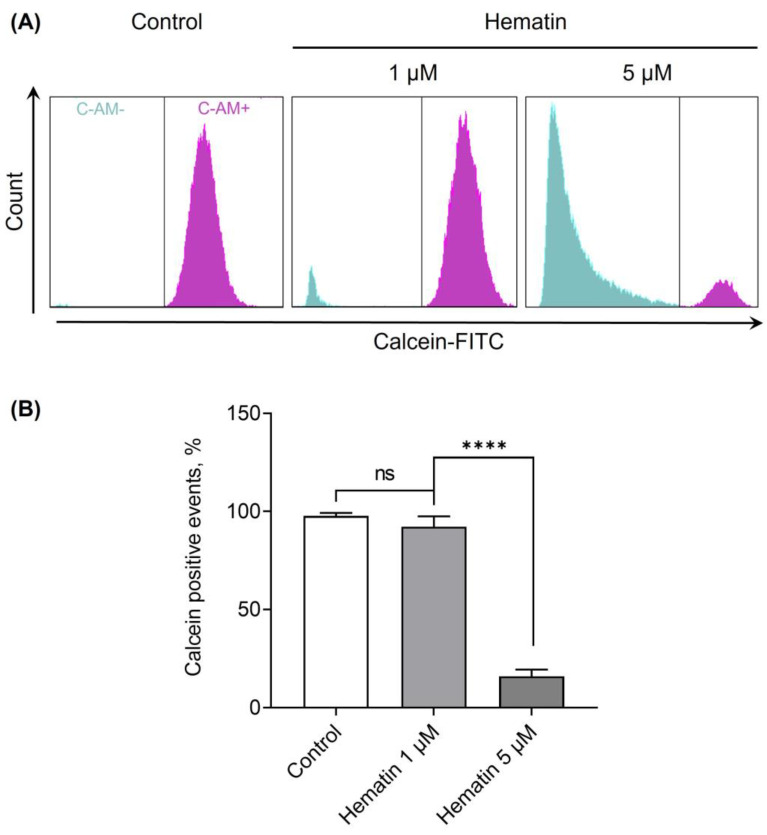
Hematin application reduced RBC viability. RBCs were incubated with calcein-AM (C-AM, 0.5 µM, 40 min, 37 °C). Hematin at indicated concentrations was added to the stained cells diluted in HEPES buffer (10^6^ cells/mL) and immediately analyzed by flow cytometry. Shown are the representative data by flow cytometry (**A**) and quantitative analysis (**B**). Data in (**B**) are presented as means ± SD. One-way ANOVA, Tukey HSD post hoc, **** *p* < 0.0001, ns—not significant, *n* = 5.

**Figure 3 cells-13-00554-f003:**
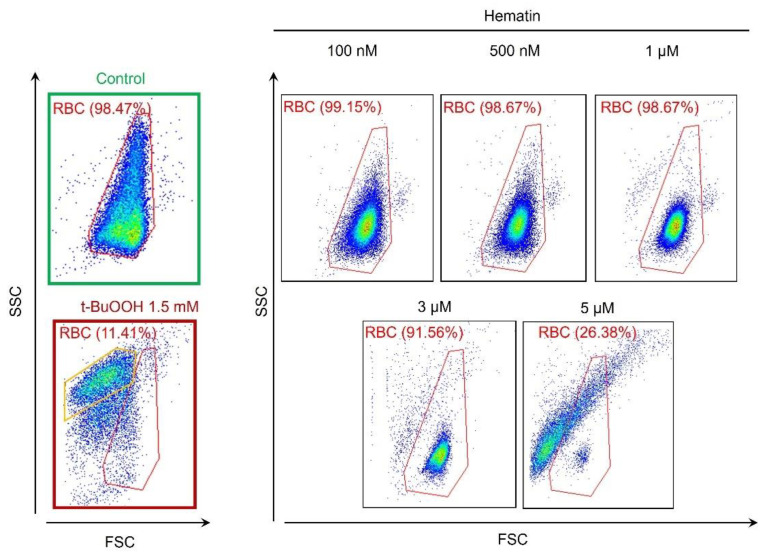
Hematin-induced spherization and hemolysis of RBCs. Hematin was added to RBCs (10^6^ cells/mL) diluted in HEPES buffer and analyzed by flow cytometry. Control RBCs were gated in red. Compared to the control (framed in green on the left), the contraction of the density plot corresponded to RBCs spherization, while moving from the control gate to the left indicated the lysis of RBCs with MPs formation. t-BuOOH (1.5 mM, framed in red on the left) was used as a control to verify whether hematin induced MP formation. The yellow gate in the t-BuOOH dot plot indicates MP formation. Shown are representative dot plots from one out of five experiments.

**Figure 4 cells-13-00554-f004:**
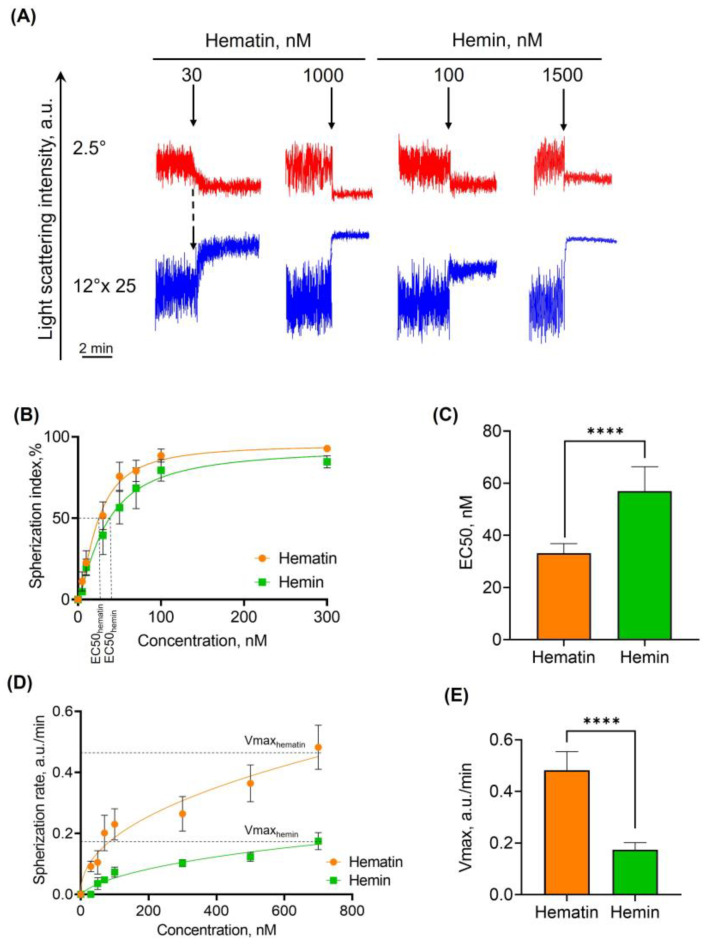
Hemin and hematin induce dose-dependent spherization of RBCs. RBCs were diluted in the HEPES buffer (10^6^ cells/mL) and added to the cuvette with a continuous stirring at 37 °C to register light scattering intensity corresponding to control. Then, hematin and hemin (black arrows) were added in the 30–1500 nM range, respectively. Narrowing of the oscillations of the laser scattering intensity (LSI) after hemin or hematin application represents the spherization of RBCs. (**A**) Representative curves from one of five experiments using the LaSca laser analyzer. For better visualization of the signal, the data of LSI at 12° were multiplied 25-fold. (**B**) Dose-dependent curves of spherization index against the concentration of hemin (orange) and hematin (green) to characterize the EC50 for both compounds. (**C**) Quantitative analysis of EC50 for spherization index. (**D**) Spherization rates of RBCs against the concentration of hemin and hematin. Spherization rates were calculated as described in the methods part. (**E**) Quantitative analysis of EC50 for maximal rates of spherization (Vmax). Data in (**B**–**E**) are presented as means ± SD. Unpaired *t*-test, **** *p* < 0.0001, *n* = 5.

**Figure 5 cells-13-00554-f005:**
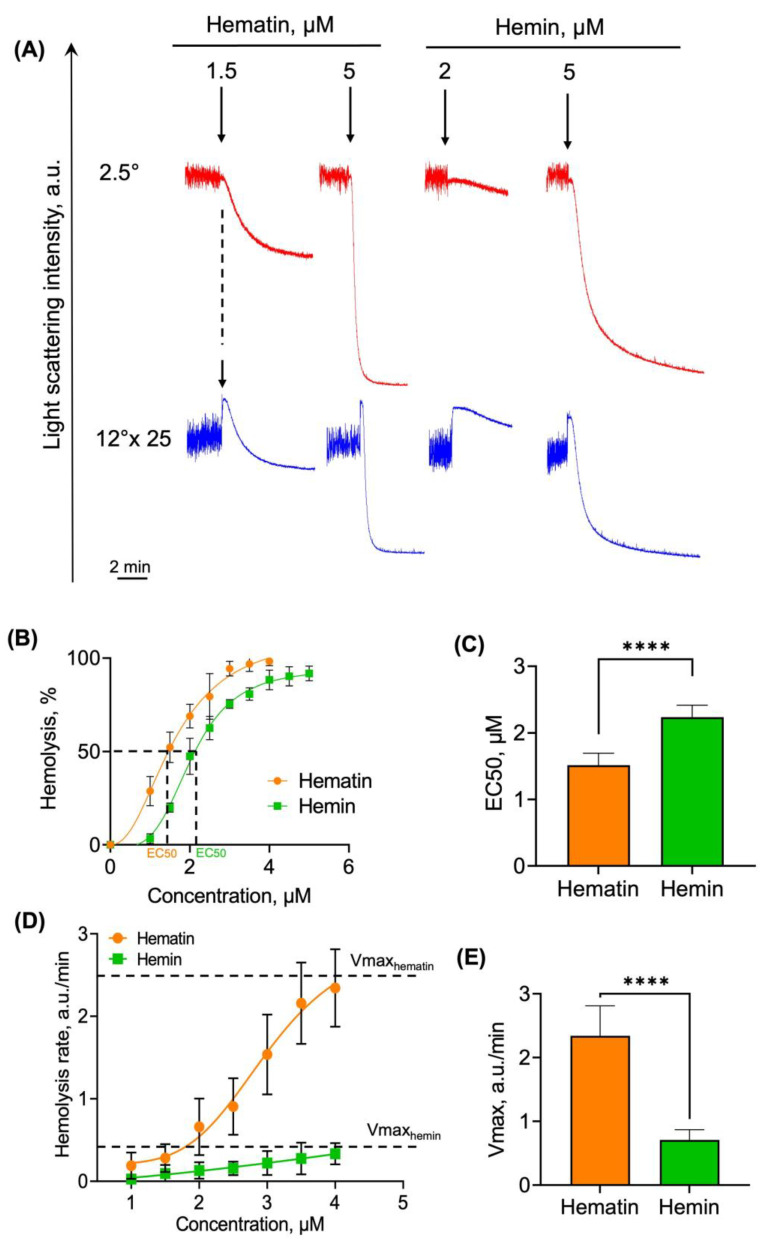
Hematin triggered the more pronounced hemolysis than hemin. RBCs were diluted in the HEPES buffer (10^6^ cells/mL) and added to the cuvette with a continuous stirring at 37 °C to register the LSI corresponding to the control. Hemin or hematin addition is indicated by the black arrows (**A**). The lysis of RBCs started after hemin and hematin addition in concentrations that exceeded the spherization-triggering concentrations. LSI descent in all angles corresponded to hemolysis of RBCs. Hemolysis rates were calculated using the original LaSca v.1498 software. (**A**) Representative curves from one of five experiments using the LaSca laser analyzer. For better visualization, the data of LSI at 12° were multiplied by 25. (**B**) Dose-dependent curves of hemolysis % plotted against the concentrations of hemin (green) and hematin (red) for EC50 characterization. (**C**) Quantitative analysis of EC50 for hemolysis percent. (**D**) Hemolysis rates against the concentration of hemin and hematin. (**E**) Quantitative analysis of maximal rates of hemolysis (Vmax) triggered by hemin and hematin. Data in (**B**–**E**) are presented as means ± SD (*n* = 5). Unpaired *t*-test, **** *p* < 0.0001, *n* = 5.

**Figure 6 cells-13-00554-f006:**
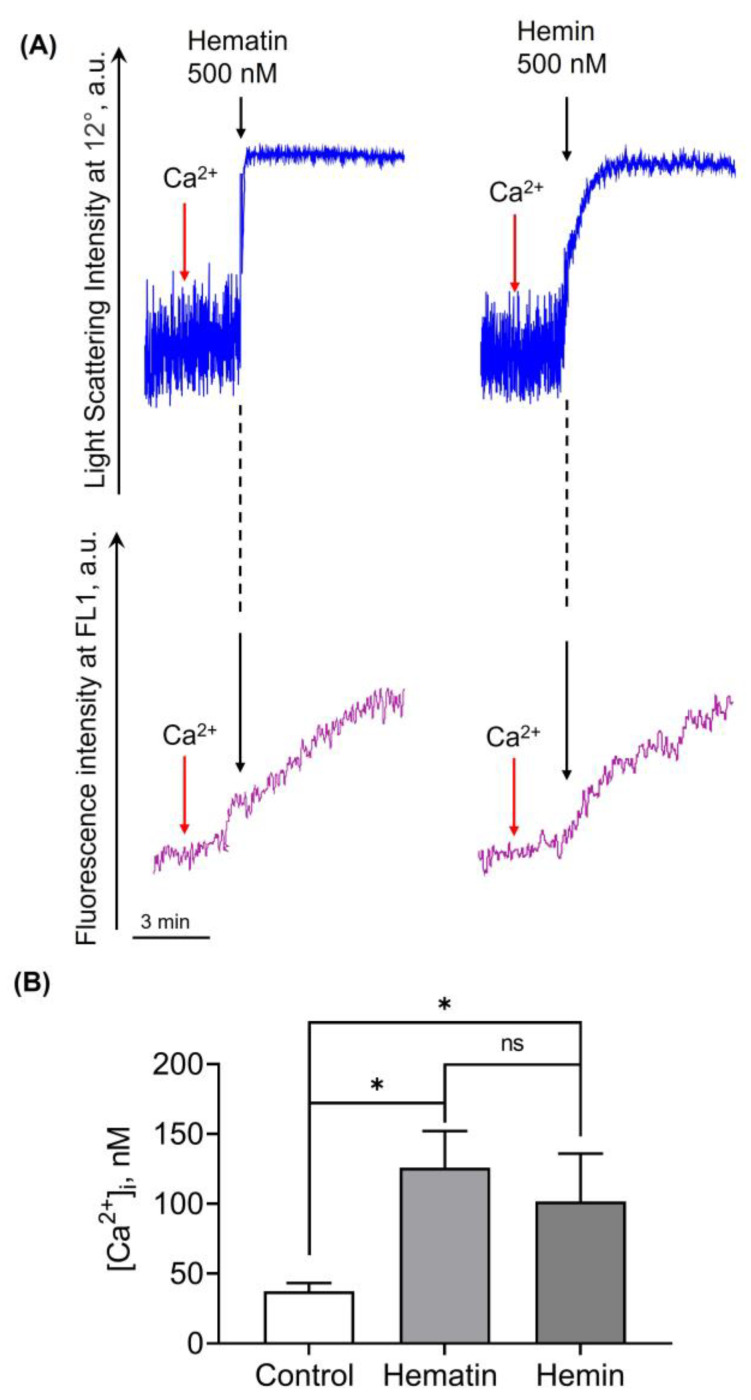
Hematin and hemin triggered the increase in intracellular calcium concentration. RBCs were incubated with Fluo-3 (10 μM, 60 min, 37 °C) and then suspended in HEPES buffer (10^6^ cells/mL) in the cuvette with continuous stirring (1200 rpm) at 37 °C and then 1.5 mM Ca^2+^ was added (A: red arrows). RBC transformation was analyzed at 12° to control RBC spherization, while Fluo-3 fluorescence intensity (FI) was registered at FL1 to visualize the corresponding changes in [Ca^2+^]_i_. First, LSI and FI for the control RBCs were registered to characterize the basal levels, then hematin or hemin at indicated concentrations were added (A: black arrows) to the cuvette with the cells, and the LSI and FI corresponding to the cell transformation were registered. (**A**) Representative curves of hemin and hematin-triggered RBC spherization (deep blue) and Fluo-3 fluorescence intensity changes (violet) registered using LaSca-TMF. (**B**) Quantitative analysis of the levels of [Ca^2+^]_i_. in absence (control) and presence of hematin or hemin. Data are presented as means ± SD. One-way ANOVA, Tukey HSD post hoc, * *p* < 0.05, ns—not significant, *n* = 5.

**Figure 7 cells-13-00554-f007:**
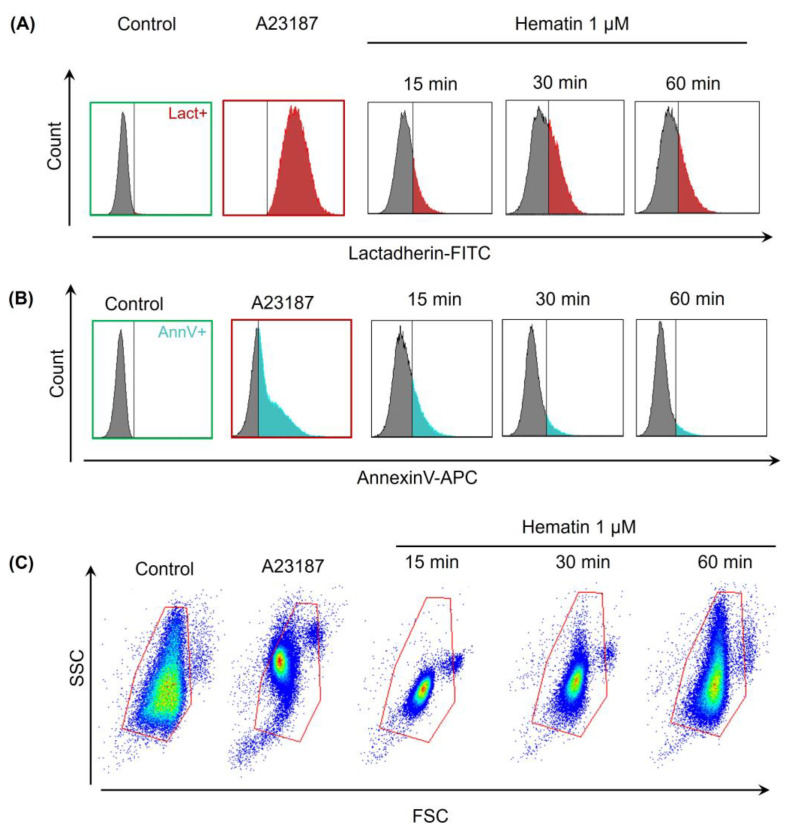
Hematin induced the acute exposure of PS on the RBC surface. RBCs (10^6^ cells/mL) were incubated with hematin (1 µM) for 15, 30, and 60 min at RT with further addition of calcium (1.5 mM). Lactadherin-FITC (**A**) and Annexin-APC (**B**) were added to the transformed cells, and after 15 min of incubation, the cells were analyzed by flow cytometry. A23187 was used as a positive control of PS exposure. RBC transformation was additionally controlled in FSC/SSC mode, and RBC distribution width returned to the levels of the control after 60 min of incubation with hematin (**C**). Shown are the data of one experiment out of five.

**Figure 8 cells-13-00554-f008:**
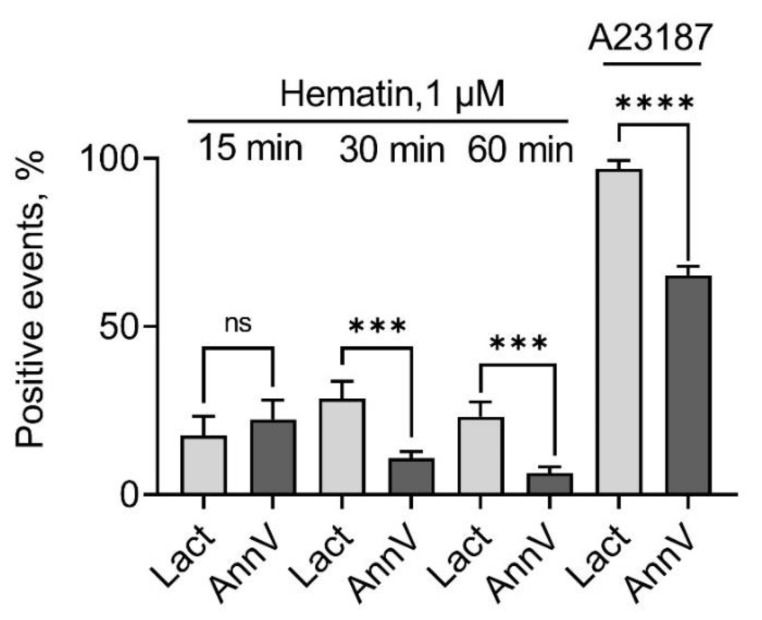
Lactadherin is a more sensitive dye for PS externalization detection. RBCs (10^6^ cells/mL) were incubated with hematin (1 µM) for 15, 30, and 60 min at RT. Then, calcium (1.5 mM) was added to the HEPES buffer. Lactadherin-FITC and Annexin-APC were added to the transformed cells, and after 15 min of incubation, the cells were analyzed by flow cytometry. Data are presented as means ± SD (*n* = 5). Unpaired *t*-test, *** *p* < 0.001, ***, *p* < 0.0001; ns, not significant.

**Figure 9 cells-13-00554-f009:**
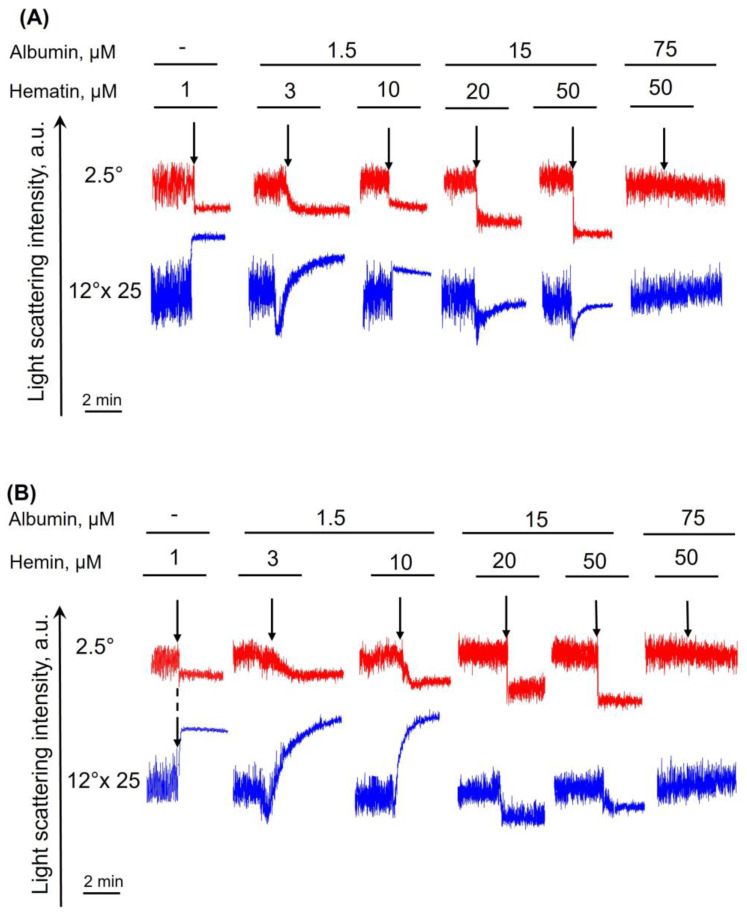
Hematin- and hemin-induced RBC spherization was abolished in the presence of albumin. RBCs (10^6^ cells/mL) suspended in the HEPES buffer with the addition of albumin at indicated concentrations were added to the cuvette with a continuous stirring at 37 °C and the LSI corresponding to control was registered. Then, hematin (**A**) or hemin (**B**) were added to the cell suspension to induce spherization of RBCs (black arrows). The narrowing of LSI oscillations reflects the spherization of RBCs. Basal LSI in the presence and absence of albumin were equal and were taken as a positive control. The LSI values at 12° were multiplied by 25 for better visualization. Shown are the original representative curves from one of five experiments on the LaSca laser analyzer.

**Figure 10 cells-13-00554-f010:**
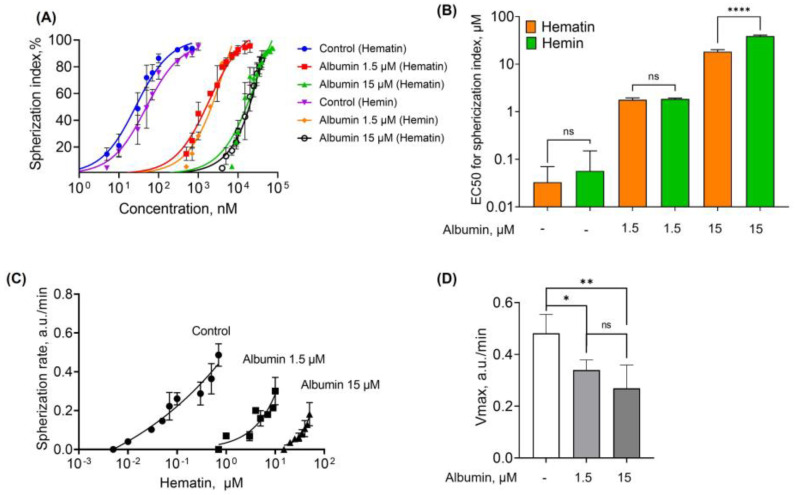
EC50 for hematin- and hemin–induced spherization increased, and V_max_ for spherization rate decreased in the presence of albumin. (**A**) The spherization index was plotted against the concentrations of hemin and hematin to characterize the EC50. (**B**) Quantitative analysis of EC50 for spherization index. (**C**) Spherization rate was plotted against the concentrations of hematin. (**D**) Quantitative analysis of maximal rates of spherization (V_max_) triggered by hematin. Data are presented as means ± SD (*n* = 5). Unpaired *t*-test, **** *p* < 0.0001; ns, not significant (**B**). One-way ANOVA, Tukey HSD post hoc, * *p* < 0.05, ** *p* < 0.01, ns—not significant (**D**).

**Figure 11 cells-13-00554-f011:**
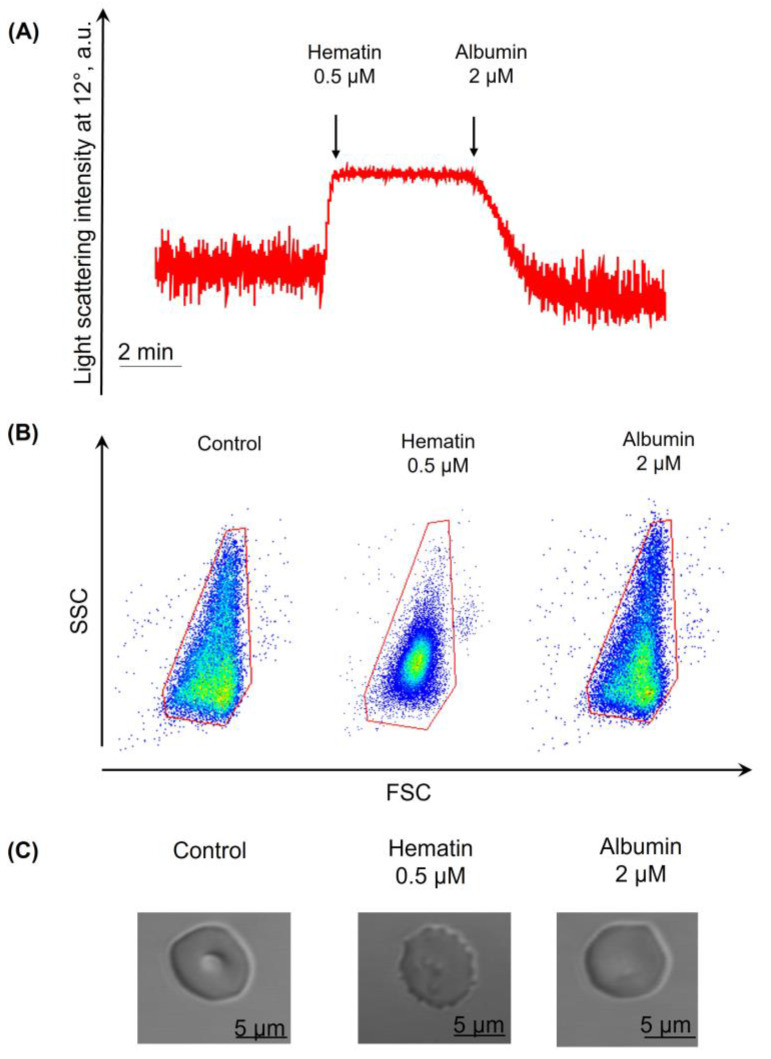
RBCs could return to the normal biconcave form after hematin application in the presence of albumin. RBCs were suspended in the HEPES buffer (10^6^ cells/mL) and added to the cuvette with a continuous stirring at 37 °C to register the LSI (**A**: red) corresponding to the control. Then, hematin (0.5 μM, A: first black arrow) was added to induce RBC spherization. Then, albumin (2 μM, A: second black arrow) was added to the transformed cells, and after 2 min, the LSI of cells (**A**) and cell distribution width within the control RBC red gate (**B**) returned to the levels of control. The sample was additionally analyzed by microscopy (**C**), which confirmed the return of the cells to their normal biconcave shape. Representative curve from the LaSca laser analyzer (**A**), dot plots, flow cytometry, (**B**), and microphotographs, one out of five experiments.

**Figure 12 cells-13-00554-f012:**
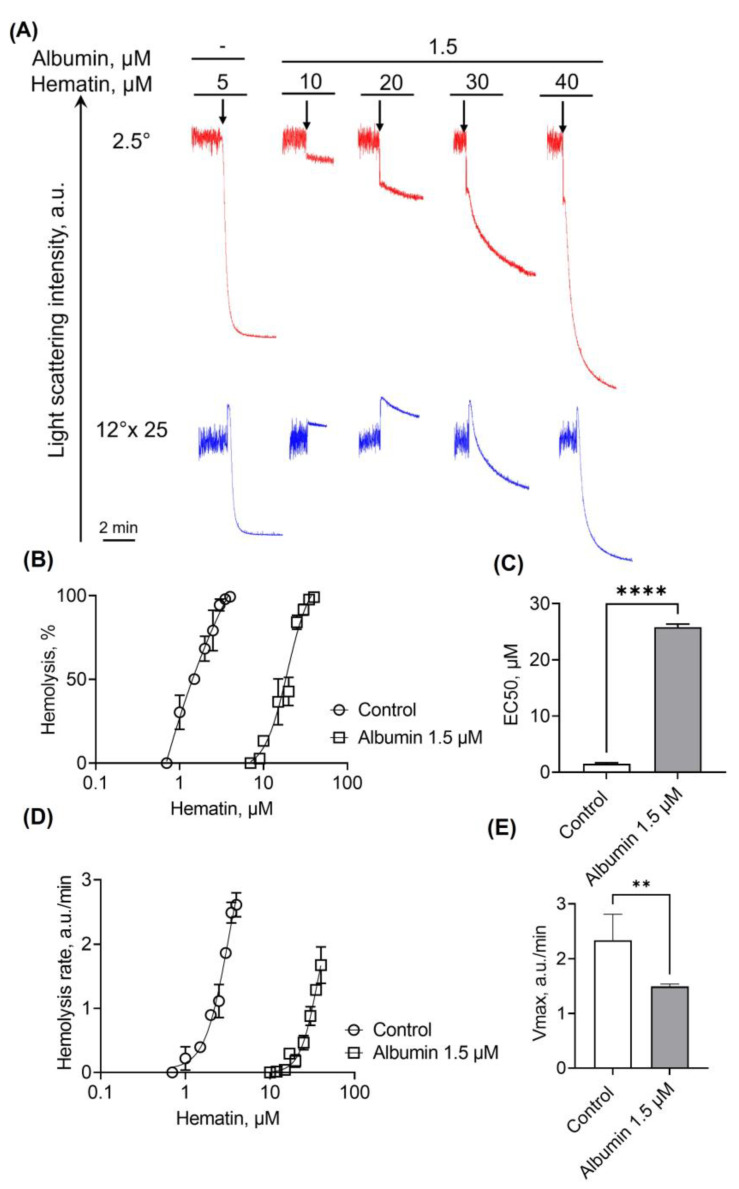
The presence of albumin diminishes hematin- and hemin-induced lysis of RBCs. RBCs (10^6^ cells/mL) were suspended in the HEPES buffer containing albumin (1.5 µM), and LSI corresponding to the control was registered. Then, hematin (black arrows) at indicated concentrations was added to the cells to induce hemolysis. The reduction of LSI in all scattering angles corresponded to hemolysis. Experiments without the addition of albumin corresponded to positive control. For better visualization, the data of LSI at 12° were enlarged 25-fold to correspond to 2.5°. (**A**) Shown are the original representative curves from one out of five experiments on the LaSca laser analyzer. (**B**) Dose-dependent curves of hemolysis against a concentration of hematin/hemin to count EC50. (**C**) Bar charts of EC50 for hematin- and hemin-induced hemolysis. (**D**) Sigmoidal curves of hemolysis rates against concentration of hematin/hemin. Hemolysis rates were calculated using the original LaSca v.1498 software. (**E**) Bar charts of maximal rates of hemolysis (Vmax). Data in (**B**–**E**) are presented as means ± SD (*n* = 5). Unpaired *t*-test, ** *p* < 0.01, **** *p* < 0.0001, *n* = 5.

**Figure 13 cells-13-00554-f013:**
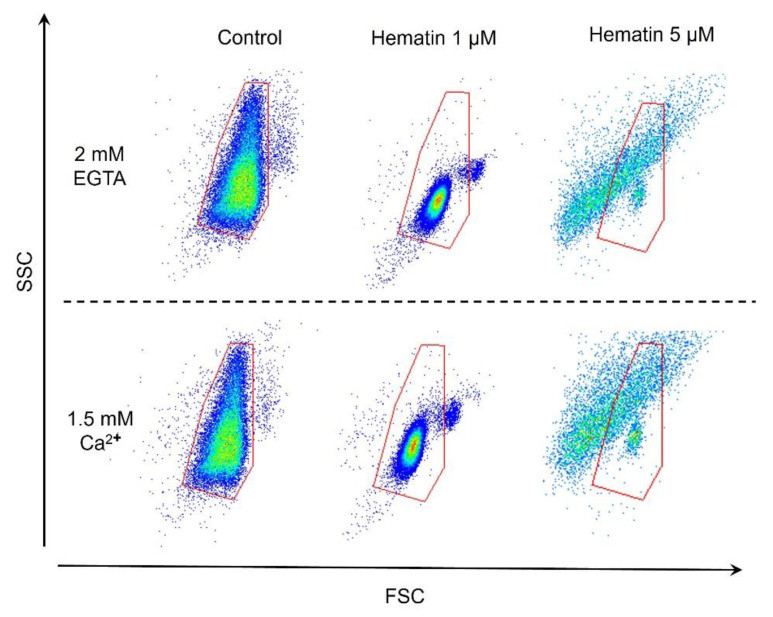
Hematin-induced transformation of RBCs is Ca^2+^-independent. RBCs were suspended in HEPES buffer (10^6^ cells/mL) in the absence (EGTA 2mM) or presence of Ca^2+^ (Ca^2+^ 1.5mM), with further analysis using flow cytometry. Red gate is a control gate for RBCs. Shown are representative dot plots from one out of five experiments.

**Table 1 cells-13-00554-t001:** Concentration ratio between hematin/hemin and albumin.

	EC50, µM	Albumin, µM	Concentration Ratio	
Hematin	1.79 ± 0.17	1.50	1.17 ± 0.11	}
Hemin	1.86 ± 0.08	1.50	1.21 ± 0.05, ns	
Hematin	18.33 ± 1.85	15.00	1.19 ± 0.12	}
Hemin	38.89 ± 2.13	15.00	2.53 ± 0.14, ****	

Data are presented as means ± SD (*n* = 5). Unpaired *t*-test, ****, *p* < 0.0001, ns, not significant.

## Data Availability

The data underlying this article will be shared at reasonable request to the corresponding author.

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
