# Peer review of "Hematin- and Hemin-Induced Spherization and Hemolysis of Human Erythrocytes Are Independent of Extracellular Calcium Concentration"

_cells, 2024, doi:10.3390/cells13060554_

Round 1

Reviewer 1 Report

Comments and Suggestions for Authors

The manuscript by Mikhailova and co-authors is devoted to the study of hematin and hemin effects on RBCs. Hematin and hemin are the oxidized forms of heme with toxic properties. The appearance of hematin/hemin in the circulating blood occurs due to the release of hemoglobin breakdown products from damaged red blood cells. This process is associated with several pathologies, such as sickle cell anemia, malaria, and thalassemia.

The above gives clinical significance to the influence of hematin/hemin on red cells' biophysical and biochemical properties. The authors demonstrate that hematin/hemin has a toxic effect on RBCs and provokes a deterioration in cells' functional properties. 

The experimental part of the study is well thought out and described in detail by the authors. The findings are discussed in detail and well described. The conclusions drawn by the authors are reasonable and appropriate.

The publication of the presented results will be helpful for a wide range of researchers and hematologists.

However, before publishing an article, I recommend that authors include two descriptive paragraphs in the Discussion section:

1. limitation of learning;

2. further directions for developing the presented results and their clinical applications.

Author Response

Dear Reviewer 1,

Thank you very much for the positive evaluation of our manuscript.

Sincerely,

On behalf of all co-authors,

Prof.Dr. Stepan Gambaryan

_________________________________________________________________________________

Comments and Suggestions for Authors

The manuscript by Mikhailova and co-authors is devoted to the study of hematin and hemin effects on RBCs. Hematin and hemin are the oxidized forms of heme with toxic properties. The appearance of hematin/hemin in the circulating blood occurs due to the release of hemoglobin breakdown products from damaged red blood cells. This process is associated with several pathologies, such as sickle cell anemia, malaria, and thalassemia.

The above gives clinical significance to the influence of hematin/hemin on red cells' biophysical and biochemical properties. The authors demonstrate that hematin/hemin has a toxic effect on RBCs and provokes a deterioration in cells' functional properties. 

The experimental part of the study is well thought out and described in detail by the authors. The findings are discussed in detail and well described. The conclusions drawn by the authors are reasonable and appropriate.

The publication of the presented results will be helpful for a wide range of researchers and hematologists.

However, before publishing an article, I recommend that authors include two descriptive paragraphs in the Discussion section:

  1. limitation of learning;
  2. further directions for developing the presented results and their clinical applications.

Answer: As suggested the separate paragraphs for limitations, future directions and clinical applications are added to the revised manuscript.

Reviewer 2 Report

Comments and Suggestions for Authors

In this manuscript, Authors separately test the effects of hematin and hemin on human red blood cells. Cell morphology and relative changes in cell size were followed by laser diffraction analusis of a stirred suspension of cells at low haematocrit (0.1%), whereas intracellular cell Ca2+ and phosphatidylserine exposure on the outside of the cell by evaluating the binding of fluorescently labelled Annexin V or lactadherin thought flow cytometry. Bright field and confocal microscopy techniques have also been adopted.

This very long manuscript exposes in a logical sequence all the experiments performed to support the conclusion that both derivatives of protoporphirin IX with liganded iron in the Fe3+ state are able to induce echinocytosis first, then spherization and finally hemolysis of red cells, with half maximal effective concentrations that are significantly lower for hematin than for hemin.

The effects, which were already known from the literature, as the Authors properly acknowledge, are indeed different in terms of EC50, although the differences are not so dramatic to justify claims of potentially different mechanisms of action or physiological role for the two compounds. In fact, there is a strong suspicion that the Authors were actually titrating their preparations of hemin/hematin solutions with red cells. The hematin solution was in fact obtained by dissolving hemin chloride, where Fe3+ is liganded to Cl- in protoporphirin IX, in sodium hydroxide, yielding the -OH-liganded Fe3+ species. There may be problems in quantifying the actual concentrations of the two derivatives spectrophotometrically, as the wavelength of maximum absorbance for hemin chloride is lower (370 nm) than that for hemin (385-400 nm) but a single wavelength of 385 nm and a single extinction coefficient was used for determining both (lines 112-120).

Major

Apart from this, the main concern is that there is no attempt to clarify the mechanism by which these compounds lead to spherization and hemolysis, and there is internal contradiction between what is stated in the premises and the conclusions drawn. More precisely: spherization could result from several mechanisms. One is that the compound intercalates into the lipid bilayer and increases the membrane permeability to Ca2+. Ca2+ entry triggers the opening of the Gardos channel with the net effect of K+/Cl- efflux, accompanied by osmotic water and cell shrinkage. This per se would not lead to spherization, unless the cell can also shed membrane by vesiculation. Moreover, even if membrane was shed as vesicles and the cell turned spherical, it would remain stable in a spherical state without hemolyzing.

The main problem with this mechanism, that the Authors invoke in the introduction as an explanation for hemin-induced spherization and hemolysis (Lines 52-62) is that it requires Ca2+ in the extracellular medium, which is not present in the HEPES buffer used for most experiments here.

The problem then is to explain spherization in a Ca2+-free medium. Here, the simplest explanation is that hemin intercalates into the lipid bilayer until, after inducing echinocytosis first, it starts making pores in the membrane leading to electrolyte equilibration at first, with the onset of a spherical morphology, and then loss of haemoglobin with the formation of ghosts.

All the data point into this direction, especially the experiment that shows that the phenomenon occurs exactly with the same sequence of events whether Ca2+ is present or absent in the medium (Figure 13).

All the other accessory experiments do not add much to the scenario: albumin in the medium increases the EC50 for both compounds with respect to shape changes and also hemolysis. It is also able to revert morphological changes and spherization evoked by hemin or hematin. However, it will be unable to revert hemolysis, although this was not tested here.

This is all expected because albumin can very efficiently sequester and lower the actual concentration of hematin/hemin in solution. Moreover, albumin is also very effective in removing hydrophobic and amphypiles from the cell membrane (for instance it is commonly used in experiments to remove the calcium ionophore A23187 and stop Ca2+ movements across the cell membrane).

For all the reasons exposed above, unless the Authors address all the above mentioned concerns and rediscuss their results accordingly, I’m not sure that the manuscript really provides any new insights into the mechanism of action of hemin in determining red cell morphological and functional changes.

Minor

Line 54: red blood cells have a membrane-skeleton, not a cytoskeleton.

Line 63: scientific publications in the literature are articles, not manuscripts.

Line 100-101: omit this line and give the buffer pH in text at the end of buffer composition.

Line 117: equation of Beer-Lambert is enough, without Law

Line 129: what is the “light scattering indicatrix of human RBCs”?

Line 136: I would not use the term “sphericity index” here for a parameter, which is dimensionless I presume, that is not well characterized in terms of physical dimensions and only quantifies relative deviation from a discoid shape towards a spherical form. Moreover, the parameter “sphericity” in the context of RBCs was defined as defined as the inverted ratio of an erythrocyte surface area to the area of a sphere of the same volume (See for instance:

Lisovskaya IL, Rozenberg JM, Yakovenko EE, Ataullakhanov FI. Maintenance of a constant area-to-volume ratio in density-fractionated human erythrocytes. Biologicheskie Membrany. 2003;20:169-177.Maybe “spherization index” is a better alternative.

Figure 1: It should be said that the experiment was done with HEPES buffer without Ca2+.

Lines 208-219 The calcein panel in Figure 1 is not commented in the text. Concerning calcein as a tool to evaluate RBC viability: calcein works as a fluorochrome whose fluorescence is strongly quenched by heme. It would not be surprising that quenching was responsible for the decrease in calcein events at the highest hemin concentrations where the haematocrit is very low (0.1%).

Line 258. Figure 3. What is the yellow gate? It is not mentioned in the figure legend.

Line 274: sphericity index again

Line 287: Hemin instead of hemin

Line 301: Hemin instead of hemin

Line 312: cytoskeleton instead of membrane skeleton

Line 315: Hemin instead of hemin

Line 351: Figure 6. It was not simply HEPES buffer but HEPES buffer PLUS Ca2+, and it is not said when Ca2+ was added or if it was there from the beginning.

Line 404: albumin in plasma is 400-600 micromolar, not mM. The “over” must be omitted.

Line 413: sphericity index again

Line 418: sphericity index again

Line 432: sphericity index again

Line 520: 500 uM/L is nonsense. It is either 500 uM or 500 umol/l.

Line 532: cytoskeleton instead of membrane skeleton

Author Response

Dear Reviewer,

We sincerely appreciate your expertise in our manuscript. We are really grateful for the time you spent on the revision. Your comments were very constructive and helped us to improve the manuscript.

Yours sincerely,

On behalf of all co-authors

Prof. Dr. Stepan Gambaryan

_____________________________________________________________________________

In this manuscript, Authors separately test the effects of hematin and hemin on human red blood cells. Cell morphology and relative changes in cell size were followed by laser diffraction analusis of a stirred suspension of cells at low haematocrit (0.1%), whereas intracellular cell Ca2+ and phosphatidylserine exposure on the outside of the cell by evaluating the binding of fluorescently labelled Annexin V or lactadherin thought flow cytometry. Bright field and confocal microscopy techniques have also been adopted.

This very long manuscript exposes in a logical sequence all the experiments performed to support the conclusion that both derivatives of protoporphirin IX with liganded iron in the Fe3+ state are able to induce echinocytosis first, then spherization and finally hemolysis of red cells, with half maximal effective concentrations that are significantly lower for hematin than for hemin.

The effects, which were already known from the literature, as the Authors properly acknowledge, are indeed different in terms of EC50, although the differences are not so dramatic to justify claims of potentially different mechanisms of action or physiological role for the two compounds. In fact, there is a strong suspicion that the Authors were actually titrating their preparations of hemin/hematin solutions with red cells. The hematin solution was in fact obtained by dissolving hemin chloride, where Fe3+ is liganded to Cl- in protoporphirin IX, in sodium hydroxide, yielding the -OH-liganded Fe3+ species. There may be problems in quantifying the actual concentrations of the two derivatives spectrophotometrically, as the wavelength of maximum absorbance for hemin chloride is lower (370 nm) than that for hemin (385-400 nm) but a single wavelength of 385 nm and a single extinction coefficient was used for determining both (lines 112-120).

Answer: Thank you very much for your critical comment. The Reviewer is correct; we did not indicate correctly how the concentrations of hemin/hematin were calculated. In the revised manuscript, we corrected accordingly: “Hematin and hemin concentrations were controlled using the molar extinction method according to the Beer-Lambert equation using the compound's millimolar extinction coefficient at two different wavelengths. For hematin ε = 58.4(mM*cm)-1 at ? = 385 nm (7). Similarly, with spectral data, we determined ε for hemin at wavelength ? = 342 nm, and ε was 59.5 (mM*cm)-1.” We also corrected the legend to Supplementary Figure 1 in the Supplementary Materials.

Major

Apart from this, the main concern is that there is no attempt to clarify the mechanism by which these compounds lead to spherization and hemolysis, and there is internal contradiction between what is stated in the premises and the conclusions drawn.

Answer: We agree with the Reviewer that in this manuscript, we did not clarify the mechanisms by which hemin and hematin lead to spherization and hemolysis. The main aim of our manuscript relates to the determination of the kinetic parameters of the process, concentrations, involvement of Ca++, and ratio of albumin to hemin and hematin. We are confident that these initial data are essential for the analysis of molecular mechanisms that trigger RBC transformation, the determination of which is our future goal.

More precisely: spherization could result from several mechanisms. One is that the compound intercalates into the lipid bilayer and increases the membrane permeability to Ca2+. Ca2+ entry triggers the opening of the Gardos channel with the net effect of K+/Cl- efflux, accompanied by osmotic water and cell shrinkage. This per se would not lead to spherization, unless the cell can also shed membrane by vesiculation. Moreover, even if membrane was shed as vesicles and the cell turned spherical, it would remain stable in a spherical state without hemolyzing.

Answer: We agree with the Reviewer that, in most cases, calcium is the main trigger of RBC spherization. However, as we showed in the manuscript, the hemin- and hematin-induced RBC transformation is calcium-independent and started before an increase of [Ca2+]i. This was one of the main findings described in our manuscript. The data presented is the basis for analyzing molecular mechanisms responsible for hemin- and hematin-induced RBC transformation, our future goal.

The main problem with this mechanism, that the Authors invoke in the introduction as an explanation for hemin-induced spherization and hemolysis (Lines 52-62) is that it requires Ca2+ in the extracellular medium, which is not present in the HEPES buffer used for most experiments here.

Answer: We are very grateful for this comment, however, as we already mentioned, we have shown that the process is independent of extracellular calcium. To make this more transparent, we added it to the introduction. “However, whether hematin- and hemin-induced transformation depend on [Ca2+]i is not known.”

The problem then is to explain spherization in a Ca2+-free medium. Here, the simplest explanation is that hemin intercalates into the lipid bilayer until, after inducing echinocytosis first, it starts making pores in the membrane leading to electrolyte equilibration at first, with the onset of a spherical morphology, and then loss of haemoglobin with the formation of ghosts.

Answer: We are very grateful for this important suggestion concerning hemin/hematin-induced RBC transformation mechanisms. We agree that this scenario might be the most realistic, however, our results do not allow us to speculate on the mechanisms. As we mentioned before, our future goal is to study distinct molecular mechanisms of calcium-independent RBC transformation.

All the data point into this direction, especially the experiment that shows that the phenomenon occurs exactly with the same sequence of events whether Ca2+ is present or absent in the medium (Figure 13).

Answer: Please see our previous reply.

All the other accessory experiments do not add much to the scenario: albumin in the medium increases the EC50 for both compounds with respect to shape changes and also hemolysis. It is also able to revert morphological changes and spherization evoked by hemin or hematin. However, it will be unable to revert hemolysis, although this was not tested here.

Answer: We partly agree that: “All the other accessory experiments do not add much to the scenario”. Yes, it was known before that albumin can prevent hemin-induced transformation of RBCs. However, our data added accurate quantitative characterization of these processes and confirmed that, according to the calculated EC50 data, one albumin molecule is binding to one hemin molecule. Additionally, for the first time, we showed that albumin could reverse hemin-induced spherization of RBCs. Concerning hemolysis, we tested it and saw that, as expected, albumin did not revert the hemolysis. We believe these data are not essential for the presented manuscript and do not include them. However, if the Reviewer persists, we can add these data.

This is all expected because albumin can very efficiently sequester and lower the actual concentration of hematin/hemin in solution. Moreover, albumin is also very effective in removing hydrophobic and amphypiles from the cell membrane (for instance it is commonly used in experiments to remove the calcium ionophore A23187 and stop Ca2+ movements across the cell membrane).

Answer: Please see our previous reply

For all the reasons exposed above, unless the Authors address all the above mentioned concerns and rediscuss their results accordingly, I’m not sure that the manuscript really provides any new insights into the mechanism of action of hemin in determining red cell morphological and functional changes.

Answer: As mentioned before, in this manuscript, we did not aim to clarify the mechanisms by which hemin and hematin trigger and lead to spherization and hemolysis. The main aim of our manuscript relates to the determination of concentrations, kinetics of the processes, involvement of Ca++, and ratio of albumin to hemin/hematin. We are confident that these initial data are essential for analyzing molecular mechanisms that trigger RBC transformation for our future work or other researchers in this field.

Minor

Line 54: red blood cells have a membrane-skeleton, not a cytoskeleton.

Answer: We thank the Reviewer for this comment, however as it is a two-dimensional structure which is basically a tethered together cytoskeleton and a lipid bilayer; and as it is known that both hematin and hemin as hydrophobic molecules easily cross the membrane layer and then stuck in the cytoskeleton, the term “cytoskeleton” was used throughout the manuscript.

Line 63: scientific publications in the literature are articles, not manuscripts.

Answer: Thank you, corrected.

Line 100-101: omit this line and give the buffer pH in text at the end of buffer composition.

Answer: Thank you, corrected.

Line 117: equation of Beer-Lambert is enough, without Law

Answer: Thank you, corrected.

Line 129: what is the “light scattering indicatrix of human RBCs”?

Answer: The corresponding information explaining the light scattering indicatrix of human RBCs is added to the Supplementary Materials.

Line 136: I would not use the term “sphericity index” here for a parameter, which is dimensionless I presume, that is not well characterized in terms of physical dimensions and only quantifies relative deviation from a discoid shape towards a spherical form. Moreover, the parameter “sphericity” in the context of RBCs was defined as defined as the inverted ratio of an erythrocyte surface area to the area of a sphere of the same volume (See for instance: 

Lisovskaya IL, Rozenberg JM, Yakovenko EE, Ataullakhanov FI. Maintenance of a constant area-to-volume ratio in density-fractionated human erythrocytes. Biologicheskie Membrany. 2003;20:169-177.Maybe “spherization index” is a better alternative.

Answer: Thank you for the commentary, we appreciate it and amended the term accordingly throughout the revised manuscript.

Figure 1: It should be said that the experiment was done with HEPES buffer without Ca2+.

Answer: Thank you, corrected.

Lines 208-219 The calcein panel in Figure 1 is not commented in the text. Concerning calcein as a tool to evaluate RBC viability: calcein works as a fluorochrome whose fluorescence is strongly quenched by heme. It would not be surprising that quenching was responsible for the decrease in calcein events at the highest hemin concentrations where the haematocrit is very low (0.1%).

Answer: We thank the Reviewer for this comment and added the following sentences in the revised manuscript. However, the fluorescence of calcein did not decrease, indicating that cell viability was maintained (Figure 1, last column). At the highest tested concentration of hematin (5 µM), we observed RBC lysis with Hb release and ghost formation (Figure 1, 5 µM) accompanied by a significant calcein fluorescence decrease.

Line 258. Figure 3. What is the yellow gate? It is not mentioned in the figure legend.

Answer: Thank you for the comment, we added the description of the yellow gate to the Figure caption in the corrected MS.

Line 274: sphericity index again

Answer: Thank you, corrected.

Line 287: Hemin instead of hemin

Answer: Thank you, we prefer to leave the names of the compounds written in lower case.

Line 301: Hemin instead of hemin

Answer: Thank you, we prefer to leave the names of the compounds written in lower case.

Line 312: cytoskeleton instead of membrane skeleton

Answer: Thank you very much, however, we prefer to use the term cytoskeleton.

Line 315: Hemin instead of hemin

Answer: Thank you, we prefer to leave the names of the compounds written in lowercase.

Line 351: Figure 6. It was not simply HEPES buffer but HEPES buffer PLUS Ca2+, and it is not said when Ca2+ was added or if it was there from the beginning.

Answer: Thank you for this critical comment. In Fig. 6 of the revised manuscript, the addition of calcium is indicated now by the red arrows.

Line 404: albumin in plasma is 400-600 micromolar, not mM. The “over” must be omitted.

Answer: Thank you, corrected.

Line 413: sphericity index again

Answer: Thank you, corrected.

Line 418: sphericity index again

Answer: Thank you, corrected.

Line 432: sphericity index again

Answer: Thank you, corrected.

Line 520: 500 uM/L is nonsense. It is either 500 uM or 500 umol/l.

Answer: Thank you for this essential remark, the mistake has been corrected.

Line 532: cytoskeleton instead of membrane skeleton

Answer: Thank you very much, however, we prefer to use the term cytoskeleton.

Reviewer 3 Report

Comments and Suggestions for Authors

In their manuscript Mikhailova et al. discuss the impact of hemin and hematin upon RBC parameters, including hemolysis, phosphatidylserine externalization and shape, all of which are altered in disease states and during RBC storage and are linked to several manifestations. The project is original and will be of interest to the scientific community and the readership of the journal, since these two heme derivatives are not generally independently studied.

I only have some minor comments that could improve the manuscript:

1) Since the authors focus on disease states when hemin and hematin are released, it would be important also to mention RBC storage, since the accumulated hemin/hematin in the supernatant could be an accelerator of storage lesions and also affect the patients post-transfusion.

2) Regarding the protective role of albumin, a recent publication showed the restoration of RBC morphology upon exposure to plasma, a finding that can further support the authors' results and conclusions (DOI: 10.3389/fphys.2022.907497).

Author Response

Dear Reviewer 3,

We are very grateful for your comments. These remarks helped us to improve the manuscript a lot.

Yours sincerely,

On behalf of all co-authors,

Prof. Dr. Stepan Gambaryan

______________________________________________________________________________

In their manuscript Mikhailova et al. discuss the impact of hemin and hematin upon RBC parameters, including hemolysis, phosphatidylserine externalization and shape, all of which are altered in disease states and during RBC storage and are linked to several manifestations. The project is original and will be of interest to the scientific community and the readership of the journal, since these two heme derivatives are not generally independently studied.

I only have some minor comments that could improve the manuscript:

1) Since the authors focus on disease states when hemin and hematin are released, it would be important also to mention RBC storage, since the accumulated hemin/hematin in the supernatant could be an accelerator of storage lesions and also affect the patients post-transfusion.

Answer: Thank you very much for this important comment. We added the corresponding information in the revised manuscript.

2) Regarding the protective role of albumin, a recent publication showed the restoration of RBC morphology upon exposure to plasma, a finding that can further support the authors' results and conclusions (DOI: 10.3389/fphys.2022.907497).

Answer: Thank you very much for this remark, this reference and explanations are added to the revised manuscript.

Round 2

Reviewer 2 Report

Comments and Suggestions for Authors

Concerning Autors' refusal to accept my suggestion and use the term "membrane skeleton" instead of "cytoskeleton", I must say that it is not a matter of taste. The two terms cannot be used indifferently.

Cytoskeleton is a much more complex tri-dimensional structure than the bidimensional membrane skleleton. It contains microfilaments (actin myosin), intermediate filaments (lamin, keratin, vimentin...) microtubules (tubulins, dynein). All these proteins are NOT PRESENT in red blood cells, because they disappear in terminal erythroid differentiation.

Therefore calling the spectrin network a cytoskeleton is a mistake.

.

Author Response

Dear Reviewer 2,

Thank you very much for that clarification and such a thorough revision of our manuscript. We are incredibly grateful as all your comments and remarks greatly improved this study.

Yours sincerely,

On behalf of all co-authors,

Prof.Dr. Stepan Gambaryan

_________________________________________________________________________________

Comments and Suggestions for Authors

Concerning Autors' refusal to accept my suggestion and use the term "membrane skeleton" instead of "cytoskeleton", I must say that it is not a matter of taste. The two terms cannot be used indifferently.

Cytoskeleton is a much more complex tri-dimensional structure than the bidimensional membrane skleleton. It contains microfilaments (actin myosin), intermediate filaments (lamin, keratin, vimentin...) microtubules (tubulins, dynein). All these proteins are NOT PRESENT in red blood cells, because they disappear in terminal erythroid differentiation.

Therefore calling the spectrin network a cytoskeleton is a mistake.

Answer: In this regard, we agree with the Reviewer, and the term is corrected throughout the manuscript (written in red and highlighted in yellow).